# Cohort Profile: CArdiovascular Risk in patients with DIAbetes in NAvarra (CARDIANA cohort)

Ibai Tamayo,[1,2,3,4] Julian Librero-Lopez,[1,2,3,4] Arkaitz Galbete,[2,3,4,5] Koldo Cambra,[4,6] Mónica Enguita-Germán,[1,2,3,4] Luis Forga,[2,7] María José Goñi,[2,7] Oscar Lecea,[3,8] Javier Gorricho,[4,9] Álvaro Olazarán,[10] Laura Arnedo,[11] Conchi Moreno-Iribas,[2,4,12,13] Javier Lafita,[3,14] Berta Ibañez-Beroiz [1,2,3,4]

For numbered affiliations see end of article.

**Correspondence to**
Dr Berta Ibáñez-Beroiz;
berta.ibanez.beroiz@navarra.es

## ABSTRACT

**Purpose** The CArdiovascular Risk in patients with DIAbetes in Navarra (CARDIANA cohort) cohort was established to assess the effects of sociodemographic and clinical variables on the risk of cardiovascular events in patients with type 1 (T1D) or type 2 (T2D) diabetes, with a special focus on socioeconomic factors, and to validate and develop cardiovascular risk models for these patients.

**Participants** The CARDIANA cohort included all patients with T1D and T2D diabetes registered in the Public Health Service of Navarra with prevalent disease on 1 January 2012. It consisted of 1067 patients with T1D (ages 2–88 years) and 33842 patients with T2D (ages 20–105 years), whose data were retrospectively extracted from the Health and Administrative System Databases.

**Findings to date** The follow-up period for wave 1 was from 1 January 2012 to 31 December 2016. During these 5 years, 9 patients (0.8%; 95% CI (0.4% to 1.6%)) in the T1D cohort developed a cardiovascular disease event, whereas for the T2D cohort, 2602 (7.7%; 95% CI (7.4% to 8.0%)) had an event. For the T2D cohort, physical activity was associated with a reduced risk of cardiovascular events, with adjusted estimated ORs equal to 0.84 (95% CI 0.66 to 1.07) for the partially active group and 0.71 (95% CI 0.56 to 0.91) for the active group, compared with patients in the non-active group.

**Future plans** The CARDIANA cohort is currently being used to assess the effect of sociodemographic risk factors on CV risk at 5 years and to externally validate cardiovascular predictive models. A second wave is being conducted in late 2022 and early 2023, to extend the follow-up other 5 years, from 1 January 2016 to 31 December 2021. Periodic data extractions are planned every 5 years.

## INTRODUCTION

Diabetes mellitus is a common metabolic disorder that affected 1-in-10 adults worldwide in 2021. Approximately, 11.5% of the total healthcare spending and 12.2% of global all-cause deaths in adults aged 20–79 years are attributable to diabetes.[1] Despite governments agreeing to halt the increase in diabetes and obesity by 2025,[2] projections for

2045 show a growth of 16% in the expected prevalence of diabetes, becoming one of the fastest growing global health emergency of the 21st century.[1]

Patients with diabetes develop common macrovascular and microvascular complications that result in an increased cardiovascular disease (CVD) risk.[3] Stratification of patients with diabetes according to their CVD risk and proper management has become an essential need for healthcare providers. However, identifying which factors and interventions impact the course of the disease is not straightforward, because their impact can differ among cohorts depending on the socioeconomic context, on the healthcare provider practices and also because of the differences in the aetiology of type 1 diabetes (T1D) and type 2 diabetes (T2D).[4 5] Focusing on this need, several cardiovascular prediction models have been proposed over the years, some of them specifically designed for patients with diabetes.[6] Choosing the CVD risk model to be applied in a particular health system is not trivial, since external validations of the models are scarce and implementation procedures are rarely straightforward.

Taking advantage of the quality of the administrative and clinical datasets in Navarra, already used for research in patients with,[7] we initiated the creation of the population-based CARDIANA (CARdiovascular risk in patients with DIAbetes in Navarra) cohort in 2016. To do so, a longitudinal extraction from multiple health and administrative databases of all patients in Navarra with T1D and T2D diabetes was conducted under the Real Word Data (RWD) framework. The baseline and first 5-year follow-up data collection ended in 2017. The aims of setting up the CARDIANA cohort were: (1) to establish a population-level dynamic cohort extraction and data integration mechanism that was nonexistent to date and could be used for research; (2) to assess which patient-level factors were determinant in the course of the disease in patients with T1D and T2D of all ages, with a particular focus on socioeconomic factors; (3) to externally validate cardiovascular risk prediction models; (4) to assess if the inclusion of socioeconomic indicators on these models improves prediction performance and (5) to quantify the impact of healthcare provider and healthcare system actions on the CVD risk of this population.

## Cohort description

The CARDIANA cohort is a population-based cohort from Navarra, an autonomous community located in a northern region of Spain with approximately 650 000 inhabitants and with a public health coverage (including both public and mixed coverage) over 99%.[8] It was designed by a multidisciplinary team involving methodologists, primary care specialists, endocrinologists, healthcare policy makers and clinical and social science researchers, many of them with expertise in the design of strategies for the management of patients with diabetes. The creation of this cohort was used as a 'case study' in the development of BARDENA, the Results Analysis Database of Navarra that is being constructed under the adoption of the Observational Medical Outcomes Partnership (OMOP) Common Data Model, which aims at harmonising electronic medical records to facilitate participation on international distributed research.

The cohort includes all users of the Public Health Service of Navarra who, as of 1 January 2012, had active codes of T1D or T2D (T89 and T90 of the International Classification of Primary Care, version 2, ICPC-2, respectively) in the Primary Care Electronic Medical Record System of Navarra (ATENEA) records. Patients with descriptions of diabetes different from T1D or T2D were excluded, as well as when severe inconsistencies in the dates of diagnosis, birth or death were found. Patients were also excluded if no registry of contact with the public health system was found either before the inclusion date and/or in the follow-up period. No other exclusion criteria were applied, and patients of all ages and conditions were considered, including patients with T2D with onset during childhood and patients with T1D with late onset during adulthood. Causes of early termination

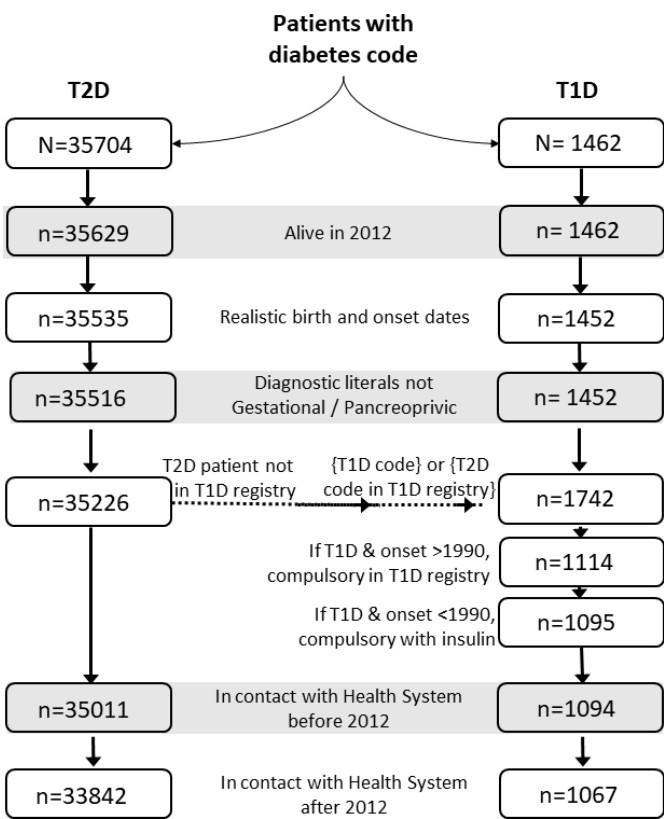

**Figure 1** Flow chart for creation of the type 1 and type 2 prevalent diabetes (T1D, T2D) cohorts.

of the patient data extraction were death or change of community/country.

Figure 1 shows the flow chart of the creation of the cohorts. The classification of patients into T1D or T2D took into account that the validity of the T2D diagnosis had been assessed in a previous study,[9] but not that of the T1D diagnosis. Hence, the ICPC-2 codes and the descriptive field that goes with the code were first used, and after that, the classification procedure was complemented with the regional registry of T1D diabetes, which was legally approved by formal order 37/2014, on 16 April[10] that includes all patients with T1D with an onset date after 1989. More precisely, we first included all patients with ICPC-2 code T89 and T90 with active Individual Health Card in ATENEA. Second, we excluded patients with incompatible or incongruent information, such as to have died before 2012, to have unrealistic birth or onset dates, or to have diagnostic literals 'gestational diabetes' or 'pancreoprivic diabetes'. Third, using the regional registry of T1D, we maintained in the T2D cohort only patients who were not in the T1D registry, and passed all patients with T2D code to the T1D cohort if they were included in the T1D registry. Fourth, we excluded from the T1D cohort all patients that had onset date >1990 but were not in the T1D registry, and also all patients with onset date <1990 that were not treated with insulin. Finally, we excluded patients that had no contact with the health system before 2012, and patients who had no contact with the health system from 2012 to 2016. Combining

the information from the health electronic records and administrative population datasets, two cohorts of prevalent patients with diabetes were created: the T1D CARDIANA cohort, with 1067 patients and the T2D CARDIANA cohort, with 33 842 patients. During the follow-up of the first wave, 33 (3.1%) patients with T1D and 455 (1.3%) patients with T2D were lost to follow-up because of having moved to another region, and information for these patients was censored at this date accordingly.

The actual follow-up period of the cohort is 5 years, from 1 January 2012 to 31 December 2016. The next data extraction process that will update longitudinal data and principal cardiovascular events is being conducted in late 2022 and early 2023, covering the period from 1 January 2017 to 31 December 2021, and further extractions are planned in 5 year waves.

### Variables, databases and integration process

Sociodemographic and clinical variables of the defined CARDIANA T1D and CARDIANA T2D cohorts came from eight clinical and administrative databases: ATENEA, LAKORA-TIS, LAMIA, HCI, HIS-LEIRE (including the Minimum Basic Data Set at hospital discharge—MBDS), the population registry, the mortality registry and the T1D registry. For future updates of the cohort, the Results Analysis Database of Navarra (BARDENA) will be used. A brief description of the original databases, which were extensively described elsewere,[11] is given in online supplemental table S1, whereas a summary of the variables considered is given in online supplemental table S2. In all, the information collected consists of all relevant structured data available from these sources generated during each contact of the patient with the health system. One set of variables was collected once and was considered fixed during the follow-up. These include the date of entry and/or exit from the health system, demographic and socioeconomic data such as the study level, lifestyle information such as tobacco use or physical activity level, the basic health zone the patients belongs to, coinsurance status,[12] baseline comorbidities and a history of cardiovascular history using the ICPC-2 codification system, among others. Some new variables were created from these previous variables, such as the Charlson weighted score[13] or the GMA[14 15] comorbidity score. The other set of variables was collected longitudinally using a time-dependent structure and included all analytical results that occurred during follow-up as well as pharmacologic treatments, health service use and fatal and nonfatal cardiovascular events. For these time-dependent variables, the date on which they occurred was also included. Cardiovascular events were considered to occur during the follow-up when CVD diagnostic codes were recorded in the mortality or the MBDS dataset, as defined in the recent SCORE2 study[16] (see the list of codes considered for fatal and non-fatal CVD in online supplemental table S3).

The integration procedure was conducted by the Statistic Institute of Navarra (NASTAT) and the Directorate-General for Informatics, Telecommunications and Innovation of the Health Department of Navarra, who supervised the data extraction and guaranteed fulfilment of the law in terms of personal data protection. Afterward, the anonymised databases were provided to the research team.

### Patient and public involvement
None.

### Findings to date
The T1D and T2D CARDIANA cohorts consisted of 1067 and 33 842 patients, respectively. Their sociodemographic characteristics are given in table 1. No adjustment has been included due to the descriptive nature of the objective, but information on both cohorts is presented in parallel. Patients in both cohorts were primarily men (57.4% and 55.7% in T1D and T2D cohorts, respectively), and only 5% were immigrants. Compared with patients in the T2D cohort, patients in the T1D cohort were much younger (mean age 36.9 years in T1D vs 69.4 years in T2D), had a higher probability of being part of the workforce (84.5% vs 26.6%), had a higher-income level (38.3% vs 27.5% had over €18 000 per year) and also higher educational attainment (17.8% vs 4.7% had university studies).

Health-related patients' status at baseline, including lifestyle data, laboratory tests values, and office measured parameters and comorbidities are given in table 2. The mean duration of diabetes was 3 years higher in patients with T1D than in patients with T2D (11.0 vs 8.1 years), but their comorbidity indices were lower, with a weighted Charlson score equal to 1.2 vs 2.1 and a weighted Adjusted Morbidity Groups (AMG, a measure of morbidity used in our health service that clasify general population according to type and complexity of the diseases they have) equal to 6.0 vs 11.4, respectively. Similarly, patients with T1D have much lower prevalence of CVD history (4.9% vs 23.8%), were more frequently active (71.5% vs 55.9%) and alcohol abstinent (69.0% vs 66.5%) but had higher probability of being smokers (32.2% vs 17.7%).

Regarding clinical and laboratory test parameters, patients with T1D showed much better control of their body mass index (25.7 vs 30.6 kg/m$^2$) but worse control of their glycosylated hemoglobin (HbA1c) levels (8.2% vs 7.1%). They also had better control of the other metabolic parameters considered, namely, high-density lipoprotein (62.0 vs 48.9 mg/dL), low-density lipoprotein (105.7 vs 111.4 mg/dL), triglycerides (84.2 vs 142.7 mg/dL) and albumin to creatinine ratio levels (13.4 vs 35.9).

The use of primary care services during the last year before baseline (table 3) was lower in patients with T1D than in patients with T2D for the total number of visits (17.1 vs 21.1 visits/year) and for all types of visits, except for emergency visits (0.6 vs 0.5 visits/year) and remote visits to nursing (4.1 vs 1.1 visits/year). Active prescriptions of antihypertensive medication were 41 points less frequent in patients with T1D (15.2% vs 56.2%), and their lipid-lowering treatment use was less than half (18.8% vs

**Table 1** Demographic and socioeconomic characteristics of the T1D and T2D CARDIANA cohorts at baseline (1 January 2012)

| | T1D CARDIANA cohort | | | T2D CARDIANA cohort | | |
|---|---|---|---|---|---|---|
| | **Male** | **Female** | **Total** | **Male** | **Female** | **Total** |
| n | 612 | 455 | 1067 | 18840 | 15002 | 33842 |
| Age, mean (SD) | 36.6 (15.9) | 37.2 (17.8) | 36.9 (16.7) | 67.1 (12.3) | 72.3 (13.0) | 69.4 (12.8) |
| Working status, n(%) | | | | | | |
| Unemployed | 36 (7.8) | 21 (6.7) | 57 (7.3) | 836 (5.0) | 609 (5.8) | 1445 (5.3) |
| Working | 392 (84.5) | 264 (84.6) | 656 (84.5) | 5506 (33.1) | 1713 (16.3) | 7219 (26.6) |
| Pensioner | 36 (7.8) | 27 (8.7) | 63 (8.1) | 10308 (61.9) | 8157 (77.8) | 18465 (68.1) |
| (Missing) | 148 | 143 | 291 | 2190 | 4523 | 6713 |
| Continent of origin, n (%) | | | | | | |
| Spain | 559 (94.6) | 415 (94.7) | 974 (94.7) | 17645 (95.9) | 13853 (94.7) | 31498 (95.3) |
| Europe | 15 (2.5) | 5 (1.1) | 20 (1.9) | 293 (1.6) | 240 (1.6) | 533 (1.6) |
| Africa | 9 (1.5) | 6 (1.4) | 15 (1.5) | 171 (0.9) | 115 (0.8) | 286 (0.9) |
| America | 8 (1.4) | 11 (2.5) | 19 (1.8) | 269 (1.5) | 399 (2.7) | 668 (2.0) |
| Asia | 0 (0.0) | 1 (0.2) | 1 (0.1) | 23 (0.1) | 29 (0.2) | 52 (0.2) |
| Australia | 0 (0.0) | 0 (0.0) | 0 (0.0) | 1 (0.0) | 0 (0.0) | 1 (0.0) |
| (Missing) | 21 | 17 | 38 | 438 | 366 | 804 |
| Copayment category, n (%) | | | | | | |
| <18000 | 338 (56.4) | 301 (68.9) | 639 (61.7) | 11445 (62.9) | 12223 (84.6) | 23668 (72.5) |
| ≥18000 | 261 (43.6) | 136 (31.1) | 397 (38.3) | 6747 (37.1) | 2229 (15.4) | 8976 (27.5) |
| (Missing) | 13 | 18 | 31 | 648 | 550 | 1198 |
| Study level, n(%) | | | | | | |
| No formal education | 132 (22.8) | 79 (18.5) | 211 (21.0) | 5245 (28.6) | 6033 (41.4) | 11278 (34.3) |
| Primary school | 226 (39.1) | 157 (36.9) | 383 (38.1) | 9744 (53.1) | 7410 (50.8) | 17154 (52.1) |
| High school | 136 (23.5) | 95 (22.3) | 231 (23.0) | 2227 (12.1) | 711 (4.9) | 2938 (8.9) |
| University level | 84 (14.5) | 95 (22.3) | 179 (17.8) | 1121 (6.1) | 424 (2.9) | 1545 (4.7) |
| (Missing) | 34 | 29 | 63 | 503 | 424 | 927 |
| Mean income, mean (SD) | 12011.6 (1803.8) | 12099.3 (1742.3) | 12048.9 (1777.6) | 11748.2 (1845.8) | 11531.6 (1739.5) | 11652.3 (1802.7) |
| Income quintile, n(%) | | | | | | |
| (7300, 10565) | 128 (21.7) | 78 (17.8) | 206 (20.0) | 3539 (19.2) | 3144 (21.5) | 6683 (20.2) |
| (10565, 11416) | 110 (18.6) | 97 (22.1) | 207 (20.1) | 3515 (19.1) | 3099 (21.2) | 6614 (20.0) |
| (11416, 12240) | 122 (20.6) | 84 (19.2) | 206 (20.0) | 3577 (19.4) | 2980 (20.4) | 6557 (19.8) |
| (12240, 13394) | 118 (20.0) | 88 (20.1) | 206 (20.0) | 3820 (20.8) | 2799 (19.1) | 6619 (20.0) |
| (13394,17708) | 113 (19.1) | 91 (20.8) | 204 (19.8) | 3949 (21.5) | 2615 (17.9) | 6564 (19.9) |
| (Missing) | 21 | 17 | 38 | 440 | 365 | 805 |

Percentage for each category are column percentages (% of patients in each category for each cohort), unless otherwise indicated (mean and SDs are given in quantitative variables).
CARDIANA, CArdiovascular Risk in patients with DIAbetes in Navarra; T1D, type 1 diabetes; T2D, type 2 diabetes.

49.3%). Similar differences were observed in antithrombotic treatment prescription (15.6% vs 38.2%), but baseline glucose-lowering treatment prescription was higher in patients with T1D (83.6% vs 67.2%).

Only nine patients (0.8%; 95% CI (0.4% to 1.6%)) in the T1D cohort developed a CVD event, five of which were fatal (0.5% of the total cohort). In the same follow-up period, 22 patients died from non-cardiovascular-related events. For the T2D cohort, 2602 (7.7%; 95% CI (7.4% to 8.0%)) had an event and 1268 of them were fatal (3.7% of the total cohort). During this follow-up, 5072 patients died from non-cardiovascular-related events.

For the T2D cohort, the occurrence of CVD events along the follow-up has been associated with physical

**Table 2** Clinical and lifestyle characteristics of the T1D and T2D CARDIANA cohorts at baseline (1 January 2012)

| | T1D CARDIANA cohort | | | T2D CARDIANA cohort | | |
|---|---|---|---|---|---|---|
| | **Male** | **Female** | **Total** | **Male** | **Female** | **Total** |
| **n** | 612 | 455 | 1067 | 18840 | 15002 | 33842 |
| Clinical parameters, mean (SD) | | | | | | |
| Duration of diabetes (years) | 10.8 (9.2) | 11.1 (9.1) | 11.0 (9.1) | 7.8 (5.8) | 8.5 (6.3) | 8.1 (6.0) |
| Body mass index (Kg/m$^2$) | 26.2 (4.5) | 25.2 (5.0) | 25.7 (4.7) | 30.2 (5.3) | 31.0 (6.4) | 30.6 (5.8) |
| Systolic blood pressure (mm Hg) | 124.0 (19.0) | 121.4 (19.9) | 122.9 (19.4) | 135.5 (16.9) | 135.7 (17.9) | 135.6 (17.3) |
| Diastolic blood pressure (mm Hg) | 72.3 (12.3) | 71.0 (9.8) | 71.7 (11.2) | 76.6 (10.5) | 75.8 (10.5) | 76.2 (10.5) |
| Laboratory tests, mean (SD) | | | | | | |
| HbA1c (%) | 8.2 (1.6) | 8.2 (1.4) | 8.2 (1.5) | 7.0 (1.3) | 7.1 (1.3) | 7.1 (1.3) |
| Fasting glucose (mg/dL) | 179.9 (97.6) | 172 (80.5) | 176.3 (89.3) | 141.6 (44.6) | 139.0 (45.4) | 140.4 (45.0) |
| Total cholesterol (mg/dL) | 188 (42.1) | 188 (31.2) | 188 (36.9) | 183.2 (39.2) | 194.0 (37.9) | 188.0 (39.0) |
| High-density lipoprotein (mg/dL) | 57.5 (17.0) | 68.4 (17.8) | 62.0 (18.2) | 46.3 (13.3) | 52.1 (14.4) | 48.9 (14.1) |
| Low-density lipoprotein (mg/dL) | 107.2 (29.8) | 103.6 (28.7) | 105.7 (29.4) | 108.9 (31.9) | 114.4 (32.4) | 111.4 (32.2) |
| Triglicerides (mg/dL) | 90.7 (58.5) | 75.1 (37.7) | 84.2 (51.4) | 141.5 (77.3) | 144.1 (68.9) | 142.7 (73.7) |
| Creatinine (mg/dL) | 0.9 (0.3) | 0.7 (0.2) | 0.8 (0.3) | 1.1 (0.6) | 0.9 (0.5) | 1.0 (0.6) |
| Albumin to creatinine ratio | 12.9 (30.9) | 14.0 (31.5) | 13.4 (31.2) | 40.2 (165.6) | 30.4 (134.0) | 35.9 (152.4) |
| Lifestyle data, n(%) | | | | | | |
| Smoking status non-smoker | 149 (49.2) | 150 (60.2) | 299 (54.2) | 6009 (37.9) | 11025 (85.1) | 17034 (59.1) |
| Ex-smoker | 51 (16.8) | 24 (9.6) | 75 (13.6) | 5810 (36.6) | 887 (6.8) | 6697 (23.2) |
| Smoker | 103 (34.0) | 75 (30.1) | 178 (32.2) | 4052 (25.5) | 1041 (8.0) | 5093 (17.7) |
| (Missing) | 309 | 206 | 515 | 2969 | 2049 | 5018 |
| Alcohol abstinent | 150 (59.5) | 159 (81.1) | 309 (69.0) | 7082 (47.5) | 10805 (90.1) | 17887 (66.5) |
| Moderate drinker | 95 (37.7) | 37 (18.9) | 132 (29.5) | 7044 (47.3) | 1137 (9.5) | 8181 (30.4) |
| Heavy drinker | 7 (2.8) | 0 (0.0) | 7 (1.6) | 777 (5.2) | 44 (0.4) | 821 (3.1) |
| (Missing) | 360 | 259 | 619 | 3937 | 3016 | 6953 |
| Physical activity inactive | 7 (4.0) | 11 (7.6) | 18 (5.6) | 1431 (9.8) | 1931 (16.1) | 3362 (12.6) |
| Partially active | 38 (21.7) | 35 (24.3) | 73 (22.9) | 3966 (27.2) | 4405 (36.8) | 8371 (31.5) |
| Active | 130 (74.3) | 98 (68.1) | 228 (71.5) | 9207 (63.0) | 5647 (47.1) | 14854 (55.9) |
| (Missing) | 437 | 311 | 748 | 4236 | 3019 | 7255 |
| Comorbidities | | | | | | |
| Charlson score, mean (SD) | 1.2 (0.8) | 1.2 (0.8) | 1.2 (0.8) | 2.2 (1.7) | 2.1 (1.6) | 2.1 (1.7) |
| *AMG score, mean (SD) | 5.5 (3.8) | 6.7 (4.7) | 6.0 (4.2) | 10.7 (5.8) | 12.2 (5.9) | 11.4 (5.9) |
| Previous CVD | 31 (5.1) | 21 (4.6) | 52 (4.9) | 4804 (25.5) | 3236 (21.6) | 8040 (23.8) |
| Diabetes-related comorbidities, n(%) | | | | | | |
| Retinopathy (%) yes | 25 (35.2) | 11 (18.6) | 36 (27.7) | 670 (15.3) | 616 (18.9) | 1286 (16.8) |
| (Missing) | 541 | 396 | 937 | 14451 | 11738 | 26189 |
| Amputation, yes | 1 (1.2) | 0 (0.0) | 1 (0.7) | 160 (2.0) | 54 (0.8) | 214 (1.5) |
| (Missing) | 532 | 390 | 922 | 10865 | 8252 | 19117 |
| Diabetic foot risk, none | 4 (5.9) | 4 (6.2) | 8 (6.0) | 230 (3.2) | 212 (3.5) | 442 (3.3) |
| Superficial ulcer | 53 (77.9) | 56 (86.2) | 109 (82.0) | 6048 (83.6) | 4809 (79.7) | 10857 (81.8) |
| Deep tissue ulcers without abcess | 8 (11.8) | 3 (4.6) | 11 (8.3) | 652 (9.0) | 704 (11.7) | 1356 (10.2) |
| Deep tissue ulcers with abcess | 2 (2.9) | 1 (1.5) | 3 (2.3) | 182 (2.5) | 244 (4.0) | 426 (3.2) |
| Localised gangrene | 1 (1.5) | 1 (1.5) | 2 (1.5) | 113 (1.6) | 62 (1.0) | 175 (1.3) |
| Extensive gangrene | 0 (0.0) | 0 (0.0) | 0 (0.0) | 10 (0.1) | 4 (0.1) | 14 (0.1) |
| (Missing) | 544 | 390 | 934 | 11605 | 8967 | 20572 |

Percentage for each category are column percentages (% of patients in each category for each cohort) unless otherwise indicated (mean and SDs are given in quantitative variables).
*AMG: adjusted morbidity groups.
CARDIANA, CArdiovascular Risk in patients with DIAbetes in Navarra ; CVD, cardiovascular disease; HbA1c, Glycosylated hemoglobin; T1D, type 1 diabetes; T2D, type 2 diabetes.

Table 3  Use of primary care services during the year previous to baseline (2011) and active prescriptions at baseline (1 January 2012)

| | T1D CARDIANA cohort | | | T2D CARDIANA cohort | | |
|---|---|---|---|---|---|---|
| | **Male** | **Female** | **Total** | **Male** | **Female** | **Total** |
| **n** | 612 | 455 | 1067 | 18840 | 15002 | 33842 |
| Total visits, mean (SD) | 15.9 (11.4) | 19.0 (12.6) | 17.1 (12.2) | 19.4 (19.1) | 23.2 (20.8) | 21.1 (19.9) |
| Visits at office, by professional, mean (SD) | | | | | | |
| Nursing | 5.2 (6.7) | 5.6 (6.9) | 5.4 (6.8) | 7.1 (10.3) | 7.6 (9.7) | 7.3 (10.1) |
| Physician | 3.8 (4.3) | 5.2 (5.5) | 4.4 (4.9) | 6.4 (5.8) | 7.3 (6.5) | 6.8 (6.1) |
| Social worker | 0.8 (1.4) | 1.3 (3.4) | 1.0 (2.5) | 1.7 (2.4) | 1.9 (3.6) | 1.7 (3.0) |
| Emergency | 0.1 (0.3) | 0.1 (0.5) | 0.1 (0.4) | 0.1 (0.6) | 0.2 (0.9) | 0.1 (0.8) |
| Other | 0.6 (1.7) | 0.7 (1.5) | 0.6 (1.6) | 0.5 (2.7) | 0.6 (1.8) | 0.5 (2.3) |
| Visits at home, by professional, mean (SD) | | | | | | |
| Nursing | 0.1 (0.8) | 0.0 (0.4) | 0.1 (0.7) | 0.7 (5.7) | 1.4 (7.8) | 1.0 (6.7) |
| Physician | 0.1 (0.3) | 0.0 (0.2) | 0.0 (0.3) | 0.3 (1.7) | 0.6 (2.2) | 0.4 (1.9) |
| Social worker | 0.0 (0.1) | 0.0 (0.2) | 0.0 (0.2) | 0.1 (1.2) | 0.2 (1.6) | 0.1 (1.4) |
| Emergency | 0.0 (0.0) | 0.0 (0.1) | 0.0 (0.0) | 0.0 (0.1) | 0.0 (0.2) | 0.0 (0.2) |
| Other | 0.0 (0.3) | 0.0 (0.1) | 0.0 (0.3) | 0.1 (0.6) | 0.1 (0.8) | 0.1 (0.7) |
| Remote visits, by professional, mean (SD) | | | | | | |
| Nursing | 3.9 (4.9) | 4.3 (5.6) | 4.1 (5.2) | 0.9 (2.3) | 1.3 (2.8) | 1.1 (2.6) |
| Physician | 1.4 (2.3) | 1.8 (2.9) | 1.5 (2.6) | 1.5 (3.1) | 2.0 (3.9) | 1.7 (3.5) |
| Social worker | 0.0 (0.1) | 0.0 (0.1) | 0.0 (0.1) | 0.0 (0.2) | 0.0 (0.2) | 0.0 (0.2) |
| Emergency | 0.0 (0.2) | 0.0 (0.2) | 0.0 (0.2) | 0.0 (0.4) | 0.1 (0.6) | 0.1 (0.5) |
| Drug treatments use, n (%) | | | | | | |
| Antihypertensive | 95 (15.5) | 67 (14.7) | 162 (15.2) | 10303 (54.7) | 8769 (58.5) | 19072 (56.4) |
| Glucose lowering | 519 (84.8) | 373 (82.0) | 892 (83.6) | 12630 (67.0) | 10099 (67.3) | 22729 (67.2) |
| Lipid lowering | 122 (19.9) | 79 (17.4) | 201 (18.8) | 9423 (50.0) | 7249 (48.3) | 16672 (49.3) |
| Antithrombotic | 67 (10.9) | 47 (10.3) | 114 (10.7) | 7722 (41.0) | 5262 (35.1) | 12984 (38.4) |

Percentage for each category are row percentages (% of patients with each treatment in each cohort), unless otherwise indicated (mean and SDs are given in quantitative variables).
CARDIANA, CArdiovascular Risk in patients with DIAbetes in Navarra; T1D, type 1 diabetes; T2D, type 2 diabetes.

activity, with estimated ORs after matching and adjusting for confounders equal to 0.84 (95% CI 0.66 to 1.07) for the partially active group and 0.71 (95% CI 0.56 to 0.91) for the active group, compared with patients in the non-active group.[17] Note that, in this study, a slightly different CVD outcome was considered.

### Strengths and limitations

The main strength of this study is that the CARDIANA cohorts integrate exhaustive clinical, socioeconomic and behavioural information from all available administrative and clinical data sources, providing a complete framework to assess the course of the disease in patients with diabetes and the factors that affect it. Especially in relation to socioeconomic variables, these cohorts have individual information on country of origin, working status, educational level and income, which are not frequently available, with most studies using area-level proxies. Another strength is

that data have been subjected to quality control procedures before-and-after database integration.

The main limitation of this study is the possible presence of bias resulting from the use of existing electronic clinical records, which may affect different methodological aspects. First, patients with T2D without a diabetes code in the ATENEA records were not included. Although the validity of the code has been satisfactory asssessed,[9] undiagnosed patients have not been included, which in Spain it has been estimated that could account for 4%–6% of the overall prevalence of 11%–14%.[18 19] Second, data completeness can be low for some variables dependent on physicians' idiosyncratic reporting procedures, such as tobacco use or physical activity and some variables that have been considered fixed may have changed along the follow-up. Although an effort has been made to complement variables with others that had a text format, information bias may be present, so imputation methods and

sensitivity analyses will be required. Third, electronic prescriptions were only fully implemented in 2014, and at baseline it is estimated that 8%–10% of the total prescriptions have not been accounted for. Fourth, patients without any contact with the regional public health system because of using exclusively private health institutions were not included. Nevertheless, it is estimated that these patients account for less than 1% in the region.[8]

## COLLABORATION

Requests for collaborative studies are welcome, on request with a description of the planned projects from berta.ibanez.beroiz@navarra.es. They will only be considered after the approval of the research ethics committee from the solicitor institution and from the Navarra health system—Osasunbidea and the NASTAT institutions—responsible for the clinical information and the population information.

### Author affiliations
[1]Unidad de Metodología, Navarrabiomed-HUN-UPNA, Pamplona, Navarra, Spain
[2]Instituto de Investigación Biomédica de Navarra (IdISNA), Pamplona, Spain
[3]Red de Invesitigación en Cronicidad, Atención Primaria y Promoción de la Salud (RICAPPS), Pamplona, Spain
[4]Red de Investigacion en Servicios de Salud en Enfermedades Cronicas (REDISSEC), Pamplona, Spain
[5]Departamento de Estadística, Universidad Pública de Navarra, Pamplona, Spain
[6]Departamento de Sanidad, Gobierno Vasco, Pamplona, Spain
[7]Servicio de Endocrinología y Nutrición - HUN, Servicio Navarro de Salud - Osasunbidea, Pamplona, Spain
[8]Atención Primaria, Servicio Navarro de Salud - Osasunbidea, Pamplona, Spain
[9]Servicio de Evaluación y Difusión de resultados en Salud, Servicio Navarro de Salud - Osasunbidea, Pamplona, Spain
[10]Servicio Tecnologías de la Salud, Departamento de Universidad, Innovación y Transformación, Pamplona, Spain
[11]Instituto de Estadística de Navarra, Pamplona, Spain
[12]Instituto de Salud Pública y Laboral de Navarra, Pamplona, Spain
[13]CIBER de Epidemiología y Salud Pública (CIBERESP), Pamplona, Spain
[14]Servicio de Efectividad y Seguridad Asistencial, Servicio Navarro de Salud - Osasunbidea, Pamplona, Spain

**Acknowledgements** The authors thank all professionals who participated in the acquisition of the data, especially Javier Baquedano, Adriana Rivero, Mª Ángeles Nuin and Alberto Jiménez-Idiazábal. The authors belong to the group RD16/0001/0014 of REDISSEC and to the group RD21/0016/0016 of RICAPPS.

**Contributors** BI and KC designed the study, researched the data and reviewed the manuscript. BI is responsible for the overall content as guarantor. IT analysed the data and wrote the manuscript. JL-L, AG and ME researched the data and reviewed the manuscript. LF and JG participated in the design of the study, in the acquisition of the T1D registry data and reviewed the manuscript. OL, JG and AO participated in the acquisition and validation of the ATENEA, LAKORA, LAMIA, HIS-Leire and HCI datasets and reviewed the manuscript. LA participated in the acquisition and validation of the population registry and reviewed the manuscript. CM-I participated in the acquisition and validation of the mortality registry database and reviewed the manuscript. JL participated in the design of the study, contributed to the interpretation of the results and reviewed the manuscript. All authors have approved this version of the manuscript.

**Funding** This work has received funding from the Instituto de Salud Carlos III through grant PI15/02196 and via the CONCEPT project (grants PI19/00154, PI19/00381). It has also received funding from REDISSEC RD16/0001/0014 and RICAPPS RD21/0016/0016, two Spanish Networks supported by Carlos III Health Institute and the European Regional Development Funding (FEDER).

**Disclaimer** All funds are public and had no role in the design of the study, analysis or writing of the paper.

**Competing interests** None declared.

**Patient and public involvement** Patients and/or the public were not involved in the design, or conduct, or reporting, or dissemination plans of this research.

**Patient consent for publication** Not applicable.

**Ethics approval** The study protocol was approved by the Ethics Committee of Clinical Research of Navarra (Project 2015/111). This Committee approved, on 19 August 2022, to update the time-window of the cohorts to 31 December 2021. This study has a retrospective nature, and data were irreversibly anonymised prior to transfer to the research team. The study was conducted according to the amended Declaration of Helsinki, Organic Law 3/2018, the General Data Protection Regulation (EU) 2016/679 and International Guidelines for Ethical Review of Epidemiological Studies.

**Provenance and peer review** Not commissioned; externally peer reviewed.

**Data availability statement** Proposals for collaborative studies to share information on data are welcome, on request with a description of the planned projects from berta.ibanez.beroiz@navarra.es. They will only be considered after the approval of the research ethics committee from the solicitor institution and from the Navarra health system—Osasunbidea and the NASTAT institutions—responsible for the clinical information and the population information.

**ORCID iD**
Berta Ibañez-Beroiz http://orcid.org/0000-0002-7797-4845

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
