## [Reviewer comments · BMJ Open]

ARTICLE DETAILS

TITLE (PROVISIONAL)	Cohort Profile: CArdiovascular Risk in patients with DIAbetes in Navarra (CARDIANA cohort)
AUTHORS	Tamayo, Ibai; Librero-Lopez, Julian; Galbete, Arkaitz; Cambra, Koldo; Enguita, Mónica; Forga, Luis; Goñi, José; Lecea, Oscar; Gorricho, Javier; Olazarán, Álvaro; Arnedo, Laura; Moreno-Iribas, Conchi; Lafita, Javier; Ibañez, Berta

VERSION 1 – REVIEW

REVIEWER	Østergaard, Helena Bleken UMC Utrecht
REVIEW RETURNED	19-Oct-2022

GENERAL COMMENTS	In this cohort profile, the authors describe their cohort established to examine the effects of sociodemographic and clinical variables on risk of cardiovascular disease in patients with type 1 or type 2 diabetes. The cohort has several strengths, the main being in my opinion the extensive amount of data on socioeconomic factors. The manuscript gives a good overview of the collected data, however I do have some comments to be improved on, please see below: 1. Abstract. The authors state that the cohort was established to “unravel” the effects of clinical variables on cardiovascular events. I think the term unravel is exaggerated, please rephrase. Furthermore, the authors state in the abstract that the cohort was established to validate and develop cardiovascular risk models, however in aims it is only specified that the cohort’s aim is to externally validate cardiovascular risk prediction models. Please rephrase in abstract.2. Introduction. In my opinion, the introduction could be more to the point, please consider revising.3. Cohort description. The cohort includes all users of the Public Health Service of Navarra, however it is not stated how much coverage of patients with diabetes in the Navarra region this is.4. Loss to follow-up. The authors state how many participants were lost to follow-up due to other reasons than mortality, please state these reasons.5. Exclusion criteria. It is not transparent exactly how the cohorts were made, also not from Figure 1. For example, why were patients excluded with no missing health system contact dates (figure 1)? And also if no contact with health system before 2012? Please revise.
------------------	--

	6. Variables. The authors refer to a study by Moulis et al. for description of the original dataset, however more detail on where the variables stem from should be included in the manuscript. Please elaborate. 7. Fixed variables. Why were these fixed (the variables mentioned are highly subject to change during follow-up as well), and were they recorded on baseline date? 8. Events. Non-cardiovascular mortality should also be reported in the dataset, especially if planning to externally validate risk prediction models that take competing risks into account. 9. In Table 2, the SCORE model is used for calculating CVD risk. However, recently the SCORE2 model has been published (Hageman et al. Eur H Journal. 2021), which is also recommended by the ESC CVD prevention guidelines (Visseren et al. Eur Heart Journal. 2021). Suggest to use the most recent risk score for predictions. 10. The manuscript should be thoroughly checked for grammar mistakes and English writing.
--	--

VERSION 1 – AUTHOR RESPONSE

Reviewer: 1

Dr. Helena Bleken Østergaard, UMC Utrecht Comments to the Author:

In this cohort profile, the authors describe their cohort established to examine the effects of sociodemographic and clinical variables on risk of cardiovascular disease in patients with type 1 or type 2 diabetes. The cohort has several strengths, the main being in my opinion the extensive amount of data on socioeconomic factors. The manuscript gives a good overview of the collected data, however I do have some comments to be improved on, please see below:

1. Abstract. The authors state that the cohort was established to “unravel” the effects of clinical variables on cardiovascular events. I think the term unravel is exaggerated, please rephrase. Furthermore, the authors state in the abstract that the cohort was established to validate and develop cardiovascular risk models, however in aims it is only specified that the cohort’s aim is to externally validate cardiovascular risk prediction models. Please rephrase in abstract.

We acknowledge the reviewer for the suggestion. We have now changed the term “unravel” for the term “assess”, which should represent better the purpose of the cohort design. One of the aims of the study was to externally validate international cardiovascular risk models. Another aim was to build over the models that better fitted the cardiovascular disease dynamic of the population of Navarra, and to analyze if the inclusion of socioeconomic variables could improve those predictions. We have now updated the aims section accordingly, by including a specific aim regarding the assessment of the inclusion of socioeconomic indicators on prediction performance. With this change, the aim of the project is more detailed.

2. Introduction. In my opinion, the introduction could be more to the point, please consider revising.

We have now reduced the introduction to better focus on the point. More specifically, we have removed the explanation of how diabetes mellitus is characterized, as we have considered not necessary. We also have removed two sentences more, one regarding diabetes evolution and the other one regarding the difficulties to choose CVD risk models, but have included one sentence regarding the magnitude of the burden of the disease and its projections. This implies that two references have also been removed.

3. Cohort description. The cohort includes all users of the Public Health Service of Navarra, however it is not stated how much coverage of patients with diabetes in the Navarra region this is.

The reviewer is right. The information was missing in the cohort description. According to data information on the National Institute of Statistics of Spain (INE)¹, the distribution of the health coverage modality in 2020 is as follows: exclusively public for 85.54% of the population, exclusively private for the 0.24% of the population, mixed for the 13.88% of the population, and other situations for the 0.33% of the population. Patients with diabetes that have public or mixed coverage are expected to be attended mainly within the public framework, because the percentage of the price of the drugs they need to pay is low in these modalities. Hence, we expect to have very few people not included in the cohort that had been diagnosed with diabetes elsewhere. That is, the representative of the sample is very high. A sentence regarding this aspect has been included in the cohort description, together with the reference (reference 8).

¹“Asistencia sanitaria. Cifras relativas,” Instituto Nacional Estadística (INE), 2018.
<https://www.ine.es/jaxi/Tabla.htm?tpx=47935&L=0>.

4. Loss to follow-up. The authors state how many participants were lost to follow-up due to other reasons than mortality, please state these reasons.

The possible reasons to be loss to follow-up in the original database (ATENEA) were to have died, to be a duplicated register, or to have moved to another region. Patients that were duplicated in ATENEA were removed before considered them candidate to be included in the cohort, so we had no individual with this cause of loss to follow-up. Patients that died along the follow up were registered and not considered a priori censored, because some future analysis may need this information to account for competing risk. Hence, the only cause of lost to follow-up we give there refers to patients that have moved to other regions (33 T1D and 455 T2D). They were left censored at the moment they were de-registered. We have added this information at the end of the ‘cohort description’ paragraph.

5. Exclusion criteria. It is not transparent exactly how the cohorts were made, also not from Figure 1. For example, why were patients excluded with no missing health system contact dates (figure 1)? And also if no contact with health system before 2012? Please revise.

Thanks to the reviewers comment we have now spotted an erratum in Figure 1. As the reviewer pointed out, under “missing health system contact dates”, where it says “No” it should say “Yes” and vice versa.

People with no contact with the health system before 2012 was removed because, for them, we had no information at baseline, other than the code T89 or T90. It was probable that these patients, in spite of having an ‘active’ health card in 2012, were not being attended by the system, perhaps for having moved to another region or country without de-registering, or for having pass to private secure without de-registering. The total number of patients removed for this motive were one T1D patient (0.09%) and 215 T2D patients (0.61%). In the same way, we obligated to have at least one contact with the health system from 2012 to 2016 to reduce the possibility of including people that was not being really followed-up in the system (again, those that having moved to another region or country without de-registering).

The whole Figure 1 has been now simplified and rearranged. We also have added some sentences regarding the step-by-step procedure of the cohort creation in the explanation of the Figure.

6. Variables. The authors refer to a study by Moulis et al. for description of the original dataset, however more detail on where the variables stem from should be included in the manuscript. Please elaborate.

We have added a Supplementary Table S1 describing the original sources. We have included a brief description on the type of information they have, the implementation date of the electronic database, and the code system used. The previous Supplementary Table S1 is now Supplementary Table S2. In future waves we will use the BARDENA Dataset, and a data resource profile document about this dataset is being under construction these weeks.

7. Fixed variables. Why were these fixed (the variables mentioned are highly subject to change during follow-up as well), and were they recorded on baseline date?

Variables that were considered fixed were demographic and socioeconomic data such as the study level, lifestyle information such as tobacco use or physical activity level, the basic health zone the patients belongs to, coinsurance status, baseline comorbidities and past history of cardiovascular history using the ICPC-2 codification system. Variables that were collected longitudinally using a time-dependent structure were all analytic results that occurred during follow-up as well as pharmacologic treatments, health service use and fatal and nonfatal cardiovascular events.

We agree in that some variables that were considered fixed would be better considered time-dependent, in an optimal cohort. However, some of these variables, such as tobacco use or physical activity level, are not as frequently recorded as other variables such as analytical parameters (which are automatically recorded with each analytic test), or treatment use (which is automatically recorded with each prescription change). Hence, we did not expect that changes in these 'fixed' variables would be as reliable as changes in the time-dependent variables considered. On the other hand, our cohort was the first one to be constructed with all these original databases, and we looked for a reasonable trade-off between feasibility/manageability and cost/effort. At the time we designed the data extraction, we considered that the effort to obtain (and use) time-dependent variables that are filled-up using routine data was worthy. However, we had not that clear if the effort to obtain (and use) variables (most of them categorical) that could change during follow-up (but for which their actualization was not guaranteed), was worthy or not, and we decided to start with this design.

As it may be considered a limitation on the cohort construction, we have added a mention on that in the limitation section.

For the following updates, which are already being designed, we will try to obtain all these data as time-dependent data. To obtain them, it has to be clearly stated the role they have on the subsequent studies, and we are working on that now. Thank you for noticing this.

8. Events. Non-cardiovascular mortality should also be reported in the dataset, especially if planning to externally validate risk prediction models that take competing risks into account.

That is true that this information is required and important to take competing risk into account. Thanks for noticing it. The non-cardiovascular related mortality has been directly addressed in the "findings to date" section of this new version of the manuscript.

Regarding events, in the review process we have realized that a reference was wrong regarding the definition of CVD codes used, and we have changed it in 'Variables, databases and integration process' section.

9. In Table 2, the SCORE model is used for calculating CVD risk. However, recently the SCORE2 model has been published (Hageman et al. Eur H Journal. 2021), which is also recommended by the ESC CVD prevention guidelines (Visseren et al. Eur Heart Journal. 2021). Suggest to use the most recent risk score for predictions.

The data information given in Table 2 is data directly derived from the original databases (or recategorization of them if it was considered beneficial for analysis or presentation purposes). The only variables that was calculated pos-hoc was the charlson index, which was not an original variable in the database, but was created taking into account the presence of the comorbidities considered in this index, and the GMA comorbidity score, which was implemented in the original dataset.

At baseline, 1st January, 2012, there were three variables directly implemented in the original electronic data (ATENEA) as DGPs (General Data of the Patients) that referred to the CVD risk. These were the FRAMINGAM risk score, the REGICOR score and the SCORE risk score. All of them were underreported. FRAMINGHAM and REGICOR had very high number of missing data (>90%). We decided to give information of the SCORE, which had lower number of missing data. Nevertheless, it had also a very high number of missing data (61.6% in the T2D cohort and 89.3% in the T1D cohort). Hence, it was used just for descriptive purposes.

We agree with the reviewer in that most recent risk score prediction models need to be used and implemented. This is precisely the key of the project: to find and validate other models that could be adapted, if necessary, to be implemented in our context. But this is not what we give in Table 2, which refer only to the original information that was found in the databases used.

We have maintained the row that refers to the SCORE in Table 2 for descriptive purposes, but we would agree if it is considered more reasonable to remove this row.

10. The manuscript should be thoroughly checked for grammar mistakes and English writing.

The manuscript was written in English by the authors and it was sent to American Journal Experts to review English-language grammar. We have gone through it again and have slightly change some sentences. We have read the manuscript again and have change some of the sentences; we hope that now the grammar mistakes are minimal.

COI statement(s):

Reviewer: 1

Competing interests of Reviewer: No competing interests.

VERSION 2 – REVIEW

REVIEWER	Østergaard, Helena Bleken UMC Utrecht
REVIEW RETURNED	21-Dec-2022

GENERAL COMMENTS	I thank the authors for having adressed my comments, for the most part satisfactorily. Two comments remain: 1. My previous comment 9. I do acknowledge that the risks estimated by SCORE were registered at baseline and thus descriptive. However, I then think that all three CVD risk scores should be reported and that in the tabel legend the above should be specified including the amount of missing data, or suggestion to remove this line. 2. On page 9 of the manuscript, it is stated that the ICD codes for assessing CVD outcomes stem from a previous article. I highly agree with uniforming ICD codes for outcomes across cohorts and studies, however perhaps the ICD codes used for the recently developed SCORE2 model (Hageman et al. Eur H Journal. 2021) would be more appropriate, since these are more recently specified and span various cohorts and geographical regions. Also, it should be specified if only primary codes were used.
------------------	--

VERSION 2 – AUTHOR RESPONSE

Reviewer: 1

Dr. Helena Bleken Østergaard, UMC Utrecht Comments to the Author:

I thank the authors for having addressed my comments, for the most part satisfactorily. Two comments remain:

1. My previous comment 9. I do acknowledge that the risks estimated by SCORE were registered at baseline and thus descriptive. However, I then think that all three CVD risk scores should be reported and that in the tabel legend the above should be specified including the amount of missing data, or suggestion to remove this line.

We understand the comment. We have decided to remove the line regarding the SCORE, since we will not use this data, and we have not described in the text the meaning of it. That is, we will not use the information of this SCORE as provided by the records in any of the analysis we will conduct, since we will calculate this and other scores from the rest of the variables. We think that, in this way, we have gain in homogeneity of the table, since know the information provided refers to variables that are provided in the original databases and that will be used as they are displayed in the table for modeling purposes.

2. On page 9 of the manuscript, it is stated that the ICD codes for assessing CVD outcomes stem from a previous article. I highly agree with uniforming ICD codes for outcomes across cohorts and studies, however perhaps the ICD codes used for the recently developed SCORE2 model (Hageman et al. Eur H Journal. 2021) would be more appropriate, since these are more recently specified and span various cohorts and geographical regions. Also, it should be specified if only primary codes were used.

The reviewer's comment make us reflect about the ICD codes considered.

For the sake of reproducibility and homogeneity, as pointed out by the reviewer, we also consider that it is important to uniform ICD codes across cohorts and studies. Taking into account that the CVD

codes used in the SCORE2 model fulfill the requirements of our study and that the SCORE2 model covers a wide range of geographical regions, this outcome definition may be more appropriate than the one we adopted, which considered the same list of codes for fatal CVD events and non-fatal CVD events. Hence, we have decided to change the outcome definition and adopt exactly the same codes provided in Hageman et al 2021 (actual ref 16).

This change in the CVD outcome definition may have some consequences in the work we have already conducted. There is one study that has already been published [ref 17], which used the previous CVD definition. Besides, we have already conducted several validation analyses of many prediction models, which have to be re-analyzed. In spite of this, we consider that the change is worthwhile. We have some changes in the manuscript regarding this change.

- We have included the new definitions in Supplementary Material (Table S3),
- We have updated the distribution of the number of fatal and non fatal CVD events for both T1D and T2D cohort,
- We have changed the reference given before for the new reference mentioning the SCORE2 study
- When describing the results of the study on Physical activity and its relation with CVD events already published [ref 17], we have added that we used a slightly different outcome definition in that case.

On the other hand, we have added in the description of the outcome in the supplementary material of the manuscript that only primary codes were used.

Thank you for the comment. We really consider that this change will modestly contribute to the homogenization of outcome definitions.

Reviewer: 1

Competing interests of Reviewer: None.